# Utilizing Polyethylene Terephthalate PET in Concrete: A Review

**DOI:** 10.3390/polym15153320

**Published:** 2023-08-07

**Authors:** Mand Kamal Askar, Yaman S. S. Al-Kamaki, Ali Hassan

**Affiliations:** 1Highways and Bridges Engineering, Technical College of Engineering, Duhok Polytechnic University, Duhok 42001, Iraq; 2Civil Engineering Department, College of Engineering, University of Duhok, Duhok 42001, Iraq

**Keywords:** plastic waste, polyethylene terephthalate, PET, green concrete, mechanical properties

## Abstract

In general, plastic waste has been growing remarkably. Numerous waste plastic products are generated by manufacturing processes, service industries, and municipal solid waste (MSW). The increase in plastic waste increases concern about the environment and how to dispose of the generated waste. Thus, recycling plastic waste becomes an alternative technique to the disposal of plastic waste in a limited landfill. One of the solutions is to use plastic waste as recycled material in concrete construction to produce what is called green concrete. This research illustrates a summary of studies that utilized polyethylene terephthalate (PET) in concrete as a volume ratio or concrete aggregate replacement. It presents data with regard to mixing design and concrete behavior when PET is used. Moreover, using PET in concrete industries may reduce environmental pollution such as the emission of carbon dioxide and plastic waste disposal problems.

## 1. Introduction

Nowadays, plastic plays a significant role in nearly every aspect of our lives. This led to an increase in the need for proper disposal management due to the huge quantity of plastic waste. The highest percentage of plastic waste is found in containers and packaging such as bottles, product packaging, cups, etc. It can also be found in building materials, furniture, etc. [1]. Since 1950, the production of plastic has increased, specifically PET, reaching 300 million tons in 2015 [2]. Moreover, even with proper disposal of these plastic materials, plastic waste requires about 400–500 years to decompose in landfills [3,4]. Hence, many researchers studied the possibility of utilizing plastic waste as recycled material in different aspects such as concrete construction, bitumen modifications, furniture, etc. [5,6]. There are several varieties of recycled plastic applications because of their mechanical properties, low density, simple processing, relatively moderate chemical resistance (in the case of thermal and electrical insulating materials), and low cost compared with other recycled materials [1].

There are two kinds of plastic. The first is thermoplastic, which can be melted and recycled in the plastic industry. Examples of thermoplastics are high-density polyethylene (HDPE), low-density polyethylene (LDPE), polyethylene terephthalate (PET), polyethylene (PE), polyethylene polystyrene (PS), polypropylene (PP), polyamide, polyoxymethylene (POM), and polytetrafluorethylene (PTFE) [7,8,9] (Figure 1). The second type is thermosetting plastic, which cannot be melted because the molecular chains are firmly bonded with meshed crosslinks; thus, it cannot be melted in the same way as thermoplastic. Examples of thermosetting plastics are melamine, silicone, epoxy resin, phenolic, unsaturated polyester, and polyurethane. Currently, these plastic wastes are either burned or buried. These procedures, however, are costly. The pollution caused by the burning process, as well as the cost of these waste management processes, can be reduced if thermosetting plastic waste can be reused [10,11]. This study illustrates most of the studies that investigated utilizing shredded PET or PET fibers in concrete and also gives the pros and cons of using PET. The study also listed the effects of PET on different aspects of concrete properties as well as the structural behavior of concrete containing PET.

## 2. Plastic Waste Properties

Properties such as tensile strength (f_t_), thermal conductivity (k), and Young’s modulus of elasticity (E) of regularly used polymers are illustrated in Table 1. The table shows that all plastic types have a lower modulus of elasticity and thermal conductivity compared to concrete components. Both fine and coarse aggregates have elastic moduli higher than PET by about 22 times, which explains why the addition of PET to the mix decreases the overall modulus of elasticity. PE, for example, has a thermal conductivity 9.1% lower than sand. Thus, an increased PE ratio in the mix leads to a decrease in the concrete’s overall thermal conductivity. Plastic, on the other hand, has a higher tensile strength than concrete components. Hence, incorporating plastic waste into concrete may improve tensile strength [2].

## 3. Polyethylene Terephthalate (PET)

PET is the most widely used thermoplastic polyester. Thus, PET should be considered for recycling. Because polyester resins are thermosetting compounds, they are often referred to simply as “polyester”. PET is a transparent polymer with excellent mechanical capabilities and dimensional stability when subjected to varying loads. PET also offers excellent gas barrier qualities and chemical resistance [14].

PET has a wide range of applications, including bottles, thermally stabilized films, and electrical components, due to the specific properties mentioned above. Another well-known application is using PET fibers in the textile industry [15]. It accounts for around 18% of total polymer production worldwide, and synthetic fibers and bottle production represent 60% of total PET demand [16]. 

## 4. PET Waste Sources

Mainly, there are three sources of PET waste. The first main source is plastic bottles, due to their higher production quantity compared with other types. Bottles have some disadvantages, such as the recycling process, label glue, unwanted additives used in production, and PET molecular weight. The second source includes foils, which have similar disadvantages to bottles. The third source is the cord from tires. This type of recycled PET has significant issues due to the rubber and metal left for disposal as a consequence of the PET recycling process. Thus, it is currently used as an alternative fuel [1]. 

From an environmental aspect, even with proper disposal of PET waste in landfills, this leads to issues related to environmental pollution. Waste PET requires about 500 years to decompose in a landfill. This is a long period, and with a rapid increase in PET production, in a few decades, there will be issues related to the availability of landfills. Another procedure for PET disposal is burning. This is also associated with environmental problems, such as pollution. Both methods of burning and burying PET have costly procedures. Hence, reusing PET in production could reduce PET disposal issues. Many researchers have investigated adding PET to concrete mixes as a PET recycling technique instead of using old disposal methods [3,4,10,11]. 

One excellent solution instead of disposing of PET is recycling plastic waste and utilizing it in asphalt binders as a modifier for road construction [17,18]. PET can also be used as reinforcing material in concrete constructions by partial replacement of fine or coarse aggregates [19,20]. These methods are regularly used to enhance the engineering properties and result in a better service life for the modified member. As a result, it contributes to achieving economic benefits and reducing environmental impacts.

## 5. Pros and Cons of Utilizing PET in Concrete

PET has recently been used in concrete mixes in a shredded or fiber format as part of an environmental solution for plastic waste [21]. Many studies have investigated the effects of PET on concrete as an additive fiber or aggregate replacement. Although PET has some advantages, it also has some drawbacks, as listed below:

PET advantages:Adding PET fibers to the concrete improves energy absorption.The ductility of concrete is significantly enhanced by the presence of PET fibers.Utilizing PET in concrete reduces post-cracks, and this is affected by PET fiber shape.PET fibers can increase the tensile, compressive, and flexural strengths of concrete if the recommended optimum dosage is used.Advantages related to the environment and PET recycling

PET disadvantages: Concrete workability is decreased significantly with the presence of PET in the concrete mix.Utilizing PET in concrete requires a concrete mix design to reach optimum results.Replacing a high ratio of fine or coarse aggregate results in a major drop in concrete strength.Adding high amounts of PET fiber to the mix results in a reduction in the overall properties of the concrete.PET fiber production is complicated and requires extensive labor.

## 6. Utilizing PET in Concrete

Many researchers have studied the effects of PET on the mechanical properties of concrete in the last two decades [5,22,23,24]. Some researchers utilized PET plastic fibers in the concrete mix to enhance the mechanical properties of the concrete (Figure 2a). This type of utilization is defined as adding PET waste as fibers to the mix with a length of 10–100 mm, a width of 1–10 mm, a thickness of 0.1–1.0 mm, and an addition ratio of 0.25–10% [25] (Figure 2b). PET can also be used as polyester fiber in a concrete mix (Figure 2c), with a length of 3–40 mm and a diameter of 20–30 μm. Adding 0.25% PET polyester can increase compressive strength by 10–20% and flexural strength by 5–15%, with a reduction in split tensile strength of about 15–30% [26,27,28].

Additionally, shredded PET of different sizes can be added to the mix to replace either fine aggregate or coarse aggregate (Figure 2b). The percentage of aggregates replaced ranges between 5 and 30% [5]. This method is used to produce green concrete rather than enhance the mechanical properties of the concrete. The biggest drawback of reusing waste plastic in concrete applications is the reduction in strength [29,30]. Many studies, on the other hand, claim to utilize 1% PET as an additive material, which may increase concrete strength by 10%. 

## 7. Properties of Concrete Containing PET

### 7.1. Fresh Properties

Workability is represented as one of the properties of fresh concrete, which is defined as the required internal work to produce fully compacted concrete [31,32]. The fresh properties of concrete may affect the physical, mechanical, and durability performances of the concrete matrix. Workability is affected by the following factors: shape, size, surface, texture, grading distribution of aggregates, w/c ratio, presence of chemicals and minerals, cement content, and climate conditions [22]. Some tests that are performed to evaluate concrete workability include the slump test by ASTM C143 [33], the Vebe test in accordance with ACI 211.3R [34] and BS EN12350:3 [35], the compacting factor test according to BS EN 12350:4 [35], and the flow table test in accordance with BS EN 12350:5 [35]. 

As the volume ratio of the plastic waste increased, concrete workability decreased. A 40% loss in workability can happen with the replacement of 15% of fine aggregate [3]. Fiber length also leads to a reduction in concrete workability [36]. The reason is that plastic waste affects the mix’s viscosity and increases its consistency. The fibers build up a mesh structure within the mix that leads to a major reduction in concrete flow, which results in a reduction in concrete workability [37,38,39,40,41,42,43,44,45]. Moreover, the PET shape also affects workability due to its sharper and non-uniform shape [46]. In general, when PET is added to the mix, this leads to a reduction in slump test results [36] (Figure 3). Slump test results can decrease from 190 mm for the control sample to 120, 80, 65, 40, and 30 mm when 0.25, 0.50, 0.75, 1.0, and 1.25% plastic waste fibers are added to the mix, respectively [42,44]. Furthermore, a study conducted by Khatab et al. [39] resulted in the same conclusion. The slump test was reduced from 120 mm for the control sample to 75 and 60 mm, respectively, when 0.25 and 0.50% plastic waste fibers were added to the mix. On the other hand, Thomas and Moosvi [43] and Rai et al. [47] reported that adding a superplasticizer to the mixture leads to an increase in workability compared to the mix without a superplasticizer. Balling and agglomeration of fibers were not detected. 

If the plastic waste is added as a partial replacement for fine or coarse aggregate, it leads to an increase in the workability of the concrete mixture [47,64,65,66]. Moreover, Al-Manaseer and Dalal [66] claimed that adding PET fiber in a limited ratio would not affect the water content of the concrete mix as PET does not absorb mixed water. This is due to the smooth surface and non-absorptive nature of the recycled plastic waste, which led to less friction between particles. On the other hand, Silva et al. [67] claimed that the workability of concrete in which fine or coarse natural aggregate was replaced by shredded PET waste bottles decreased when coarse or fine plastic aggregates were added. Plastic fiber also generates a gap in the concrete matrix between cement and natural aggregates that results in a delay in the initial reaction between them. Adding 15% PET can lead to the segregation of concrete, and it could be because of the high w/c ratio [46]. 

### 7.2. Fresh and Dry Density

Density is defined as the weight of the volume. As concrete consists of different components such as cement, fine and coarse aggregates, water, and admixtures, changes in mix design or partial replacement of fine or coarse aggregate result in changes in concrete density [68].

Fresh concrete density is the density of concrete at the plastic stage. The fresh density of concrete containing PET is reduced when PET is added (Figure 4). This is because of the low specific gravity of PET compared to the specific gravity of natural fine or coarse aggregate [13,47,52,67,69]. Ismail and Al-Hashmi [70] agreed with the previous conclusion after testing samples containing 10%, 15%, and 20% PET, and they found that fresh density is reduced by 5%, 7%, and 8.7%, respectively. 

The density of concrete is reduced by increasing PET volume [3,47,50]. A study conducted by Hannawi et al. [71] indicated that replacing 50% of fine aggregate with PET decreased dry density to 19%. This is due to the low specific gravity of plastics compared to fine aggregate [36]. Moreover, reducing PET size while keeping the same fraction leads to a reduction in the bulk density of concrete [72]. 

**Figure 4 polymers-15-03320-f004:**
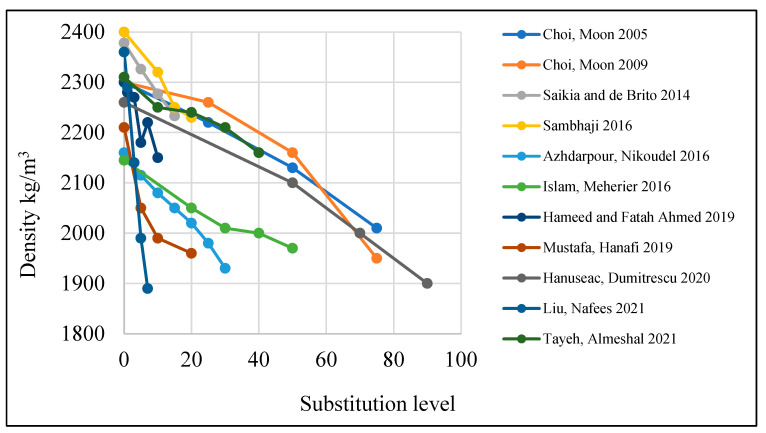
Effects of PET substitution on the dry density of concrete [48,50,52,53,54,55,59,63,73,74,75].

### 7.3. Water Absorption

Water absorption is one of the concrete features used to check the quality of concrete, and it can be used to assess concrete porosity. The water absorption and permeability of concrete are affected by the water absorption of the concrete component. Meena et al. [58] claimed that the water absorption of PET, fine aggregates, and coarse aggregates is 0%, 1.54%, and 0.85–1.1%, respectively. As permeability or water absorption is reduced, concrete will be more durable [56,76]. Won et al. [77] claimed that the permeability of concrete is reduced when a 1% volume fraction of PET is added to the concrete mix. Furthermore, partial replacement of 3% fine aggregate with PET leads to a reduction in concrete permeability and porosity [71]. The maximum amount of PET partial replacement, as claimed by Nassani et al. [78], should not exceed 5%. Adding more than 5% may increase permeability and reduce strength. Replacing 20% of fine aggregate with PET results in a 55% increase in permeability despite the effects of the superplasticizer [46]. This finding is also agreed upon by [45,71,79,80,81] (Table 2).

### 7.4. Ultrasonic Pulse Velocity

The ultrasonic pulse velocity (UPV) test is considered a nondestructive in-situ test that is usually used to evaluate the quality of concrete (Figure 5). The ASTM C597-09 Standard Test Method for Pulse Velocity Through Concrete [82] is used to measure ultrasonic wave velocity. This occurs by determining the speed of an ultrasonic pulse as it passes through a concrete member [83,84]. Slower velocities may suggest concrete with many fractures or voids, whereas higher velocities indicate good quality and continuity of the material [85]. The transducers are put on opposite sides of the material after calibration to a standard sample of the material with known properties. A simple formula (Equation (1)) can be used to calculate pulse velocity [85,86]: (1)$$Pulse\;Velosity=\frac{Width\;of\;structure}{Time\;taken\;by\;pulse\;to\;go\;through}$$

PET aggregate replacement leads to a noticeable ultrasonic pulse velocity loss [3,36,45,51,73,87] (Figure 6a). A study conducted by M. Nikbin et al. [88] claimed that the loss of ultrasonic wave velocity in samples containing more PET could be because concrete containing PET particles has a higher capacity to resist internal pressure induced by cement paste expansion. 

A researcher studied the effects of PET fibers on pulse velocity. Different waste PET fiber ratios were used: 0.25, 0.50, 0.75, 1.0, 1.25, and 1.50%. The result showed that as the PET ratio increased, pulse velocity decreased [38]. The same finding was observed by [89,90,91]. This outcome is debatable because waste PET fibers increased porosity and decreased the concrete mixture’s unit weight [3,22]. On the other hand, another research study claimed that PET did not significantly affect pulse velocity, especially over a short period of time. At 28 days, the result showed a small increase of 0.3 and 0.33% for 0.25 and 0.5% PET fibers, respectively [42]. The same finding was observed by [92], with a different result if more than 0.5% PET is added to the mixture. Results showed that there is a slight reduction in pulse velocity beyond 0.5% waste PET fibers.

### 7.5. Modulus of Elasticity

The stiffness of concrete is measured by its modulus of elasticity, which is an excellent indicator of its strength. The concrete can withstand more stress and becomes brittle as the modulus of elasticity increases. The elastic modulus of concrete is generally between 30 and 50 GPa [93]. Based on the stress–strain curve, the modulus of elasticity is calculated in accordance with ASTM C-469 [45,94]. As shown in Equation (2): (2)$$E=\frac{\left(\sigma_{2}-\sigma_{1}\right)}{e_{2}-50\;\times \;10^{-6}}$$
where *σ*_2_ is the stress that corresponds to 40% of the maximum load; *σ*_1_ is the stress that corresponds to the longitudinal strain (50 × 10^−6^); and e_2_ is the longitudinal strain produced by *σ*_2_.

The modulus of elasticity of concrete is reduced in the presence of waste PET. It is a reverse relation; when the ratio of the substituted or added PET is increased, it accompanies a reduction in the modulus of elasticity [3,45,53,67] (Figure 6b). By replacing 10% of the fine aggregate with waste PET, although there is no change in the strength of the concrete, there is a reduction in the modulus of elasticity. However, the fact that waste PET particles can be used to make concrete with a more ductile behavior is a desirable outcome [3]. The modulus of elasticity can drop from 27.2 GPa to 21.1 GPa, about 22% lower, when 20% of waste PET is replaced with fine aggregate. The drop rate in the modulus of elasticity is reduced with the reduction in the PET ratio [45]. 

### 7.6. Effects of PET on the Microstructure of Concrete

To investigate the microstructure of concrete, a scanning electron microscope (SEM) is usually used. Concrete containing PET shows a relatively irregular form that leads to the formation of pores of about 2–4 µm. Multiple bright inclusions (cement formations) encircled by hydrating agents could be observed on the surface, which improves the bonding between the PET fibers and the matrix (Figure 7). Concrete containing PET probably has a much denser interface between the PET aggregates and the cement matrix. Moreover, microcracks reduce with the presence of PET fibers [95,96]. Aslani 2019 [97] and Hou 2019 [98] reported that the compressive strength decreases with the addition of plastic fibers. Furthermore, Aslani 2019 [97] found that increasing the volume fraction of plastic fibers from 0.1% to 0.2% decreases the compressive strength by about 20%.

On the other hand, Faraj 2020 [98] claimed that concrete microstructures show improvements in compressive strength due to the distribution of the fibers within the microstructures. This leads to a reduction in the pores inside the concrete matrix. The length of the fibers has a slight influence on the compressive strength of concrete [99]. 

### 7.7. Compressive Strength

In concrete structures, compressive strength is considered one of the most essential mechanical properties, and it usually indicates the quality of the concrete [31,100]. ASTM C39 [101] is used to conduct the compressive strength tests for cylindrical concrete specimens. BS EN 12390:3 [102] is also used to find the compressive strength of concrete specimens. In general, adding PET to the concrete mix leads to a reduction in the concrete’s compressive strength, split tensile strength, modulus of elasticity, and unit weight [46,87]. Moreover, Pereira et al. [103] studied the effects of fiber volume and length on the compressive strength of concrete, and it was found that compressive strength is affected only by PET volume and is reduced when the PET ratio is increased. The reason behind it could be a consequence of the reduction in binding between cement paste and the aggregate when PET is used. Nevertheless, a 12.5% aggregate replacement rate led to considerable improvements in compressive, splitting tensile, and flexural strength (by 43, 27, and 30%, respectively) [45]. 

Belmokaddem et al., 2016 [87] conducted an experimental study and found that replacing natural aggregate results in a significant loss in compressive strength, dynamic modulus of elasticity, and ultrasonic pulse velocity with increasing ductility. On the other hand, the investigation discovered significant improvements in thermal insulation, with the concrete containing 75% PVC waste achieving a 67% reduction in thermal conductivity.

The reduction in concrete strength is due to the fact that PET particle usage causes some deficiencies in the inner structure of the concrete, resulting in a reduction in tensile strength and stiffness. This behavior could be advantageous when ductility is required [87]. Table 3 lists studies that investigated the effects of PET on the compressive strength of concrete. Moreover, Figure 8 shows that adding PET as an additional material to the concrete mix increases compressive strength if the addition ratio does not go beyond 0.4%. Where PET is used as a replacement material, the optimum ratio is 1% for fine and coarse aggregate replacement (Figure 9 and Figure 10).

### 7.8. Splitting Tensile Strength

Tensile strength is an important property of concrete because structural loads expose it to tensile cracking. In general, concrete’s tensile strength is significantly lower than its compressive strength. Concrete’s tensile strength is estimated to be around 10% of its compressive strength. Due to the difficulty of the direct method, indirect methods are used to determine tensile strength. It is worth noting that the results from these methods are higher than the results from the uniaxial tensile test. The split cylinder test and the flexural test are two indirect techniques [110].

The concrete tensile efficiency was shown to be influenced by the synergistic effect between the fiber volume and fiber length. A study conducted by Pereira et al., 2017 [103] shows that concrete with 10% fine aggregate replaced with PET particles has the same strength compared to the control sample and a lower modulus of elasticity. In other words, concrete with more ductility can be achieved with the same strength if PET is used as a fine aggregate replacement. The authors of [3] studied the effects of replacing up to 15% of PET with two water cement ratios of 0.42 and 0.54, and the result indicated that the unit weight of concrete decreased by 3.1%. The study also claimed that waste PET can be reused as a fine aggregate replacement and could enhance the mechanical properties of concrete as part of the environmental solution for waste PET. This conclusion is agreed upon by [52], with a reduction in water absorption when PET is used as a waste material substitution. 

Table 3 lists studies that investigated the effects of PET on the split tensile strength of concrete. Moreover, Figure 11 shows that adding PET as an additional material to the concrete mix would increase split tensile strength by 10–20% when a 0.4–1% PET ratio is used. In the case of using PET as a partial replacement for fine aggregate, adding 1–8% would increase split tensile strength by 1–20% (Figure 12). However, if PET is used as a coarse aggregate replacement, that would negatively affect the split tensile strength (Figure 13).

### 7.9. Flexural Strength

Flexural strength, also known as modulus of rupture, is defined as the material stress prior to yielding in a flexure test. Flexural strength is considered one of the significant properties of concrete to determine tensile strength based on bottom fiber maximum stress. The flexural strength of concrete is affected when PET is added or replaced. When replacing fine aggregate with only 5% PET with a w/c ratio between 0.5 and 0.6.5, it can increase flexural strength by 6–8%. In contrast, replacing fine aggregate with 15% PET can reduce flexural strength by 6–14%, depending on the w/c ratio [3]. Another study conducted by Dawood et al. [45] claimed that there are three main classes of replacing aggregate with PET: 0–5%, 6–15%, and 15–20%. In the first class, the flexural strength was significantly enhanced. In classes two and three, there was a gradual increase in flexural strength with the increase in the PET ratio. This conclusion is agreed upon by [3,51,70,73,111,112].

Table 3 lists studies that investigated the effects of PET on the flexural strength of concrete. Moreover, Figure 14 shows that by adding PET as an additional material to the concrete mix, no remarkable enhancement to the concrete’s flexural strength was noticed, apart from several authors who claimed a different point of view. On the other hand, Figure 15 shows that adding PET as a replacement for fine aggregate increases flexural strength by 40% when a 0.5–6% ratio is used. In the case of PET being used as a coarse aggregate replacement, it negatively affects the flexural strength (Figure 16).

**Table 3 polymers-15-03320-t003:** Effects of PET on concrete strength.

Author	Sample ID	Parameter/Remarks	F’c (Mpa)	Ft (Mpa)	Flexural (Mpa)	Slump Test (cm)	Dry Density (kg/m^3^)	Material Types	Dimension L × W × T (mm)	Ratio % V	Replacement/Addition
Choi, Moon [48]	53P0	w/c: 0.53 SP: 0.3%	31.5			10	2300	Crushed PET		0	Replacing by volume fine aggregate
	53P25		29.7			15.3	2220			25	
	53P50		26.3			19.9	2130			50	
	53P75		21.8			22.3	2010			75	
	49P0	w/c: 0.49 SP: 0.3%	34.6			10.5	2300			0	
	49P25		33.7			15.4	2230			25	
	49P50		29.1			18.0	2120			50	
	49P75		23.2			21.4	2000			75	
	45P0	w/c: 0.45 SP: 0.3%	37.2			13.5	2300			0	
	45P25		33.8			16.9	2260			25	
	45P50		31.8			18.4	2160			50	
	45P75		24.9			20.5	1940			75	
Ochi, Okubo [49]	C11	Cement: 334 kg Fine agg. 973 kg Coarse agg. 743 kg Water 217 L w/c 0.65Cement: 334 kg Fine agg. 973 kg Coarse agg. 743 kg Water 217 L	32.1		3.82	16.5		PET	30 mm with 15 mm max aggregate size	0.0	Adding as volumetric ratio
	C12		31.4		3.72	16.0				0.5	
	C13		34.8		4.12	3.5				1.0	
	C14		34.1		4.80	4.0				1.5	
	C21		34.8		4.12	9.5				0.0	
	C22		34.8		3.97					0.5	
	C23		39.6		4.21					1.0	
	C24		38.8		5.29					1.5	
	C31	w/c 0.60 Cement: 334 kg Fine agg. 973 kg Coarse agg. 743 kg Water 217 L w/c 0.55	45.1		4.21	7.0				0.0	
	C32		45.6		4.41					0.5	
	C33		47.8		4.85					1.0	
	C34		43.7		5.73					1.5	
Gupta, Rao [26]	1		49.6	3.39	4.7			PET polyester fiber	6 mm length × 0.0445 diameter	0.0	Adding as volumetric ratio
	2		59.8	-	4.5					0.2	
	3		60.0	2.23	5.0					0.25	
	4		48.0	-	4.4					1.0	
Choi, Moon [50]	W/C53	Cement: 336 kg Fine agg. 844 kg Coarse agg. 930 kg Water 178 L SP 1.008 kg/m^3^	32.1	3.3		10	2300	Shredded PET	5–15	0	Fine aggregate replacing
	W/C53		30.2	2.8		15.3	2260			25	
	W/C53		26.8	2.4		19.9	2160			50	
	W/C53		22.4	2.0		22.3	1950			75	
	W/C49	Cement: 367 kg Fine agg. 805 kg Coarse agg. 939 kg Water 180 L SP 1.101 kg/m^3^	36.4	3.0		10.5	2300			0	
	W/C49		35.3	2.8		15.4	2230			25	
	W/C49		30.3	2.4		18	2110			50	
	W/C49		24.4	2.0		21.4	2000			75	
	W/C45	Cement: 402 kg Fine agg. 771 kg Coarse agg. 941 kg Water 181 L SP 1.206 kg/m^3^	38.0	3.0		13.5	2300			0	
	W/C45		34.0	2.8		16.9	2220			25	
	W/C45		32.3	2.4		18.4	2130			50	
	W/C45		27.7	2.0		20.5	2000			75	
Albano, Camacho [51]	C0	Cement: 19.1 kg Fine agg. 68.6 kg Coarse agg. 43.6 kg w/c: 0.6	28.0	23.3		7.8		Shredded PET	-	0	Replacing by volume fine aggregate
	CS		22.9	16.9		4.0			22.3 mm	10	
	CS6		22.5	21.7		5.2			22.3/33.4 mm	10	
	CB6		22.1	22.2		3.0			33.4 mm	10	
	CSW		17.5	14.5		2.2			22.3 mm	20	
	CSBW		18.5	17.3		1.9			22.3/33.4 mm	20	
	CBW6		14.1	14.5		0.0			33.4 mm	20	
	S15	Cement: 24.1 kg Fine agg. 64.9 kg Coarse agg. 41.2 kg w/c: 0.5	21.4	28.0		8.6			-	0	
	SB5		16.9	24.4		6.5			22.3 mm	10	
	B15		18.5	25.1		7.5			22.3/33.4 mm	10	
	S25		14.3	23.6		7.0			33.4 mm	10	
	CSB5		13.3	18.8		4.2			22.3 mm	20	
	CB20		12.9	21.8		4.7			22.3/33.4 mm	20	
			9.1	19.0		0.0			33.4 mm	20	
Ramadevi and Manju [109]	C0	Cement: 425.78 kg Fine agg. 516.05 kg Coarse agg. 1175.92 kg w/c: 0.45	31	1.88	3.2			Shredded PET		0	Replacing by volume fine aggregate
	C0.5		33.1	1.99	4.4					0.5	
	C1		40.1	2.04	5.2					1	
	C2		39.8	2.11	5.7					2	
	C4		38.7	2.07	5.9					4	
	C6		38.1	2.04	5.9					6	
Pelisser, Montedo [28]	1	1:2.3:2.7:0.62	29.2		3.75	10.0		PET polyester fiber	10 mm length × 25–30 μm diameter	0.0	Adding as volumetric ratio
	2		28.3		3.6	15.5				0.05	
	3		27.0		4.2	7.0				0.18	
	4		29.5		4.4	5.0				0.30	
	5		28.3		4.23	15.5			15 mm length × 25–30 μm diameter	0.05	
	6		27.0		4.2	7.0				0.18	
	7		29.5		4.5	5.0			20 mm length × 25–30 μm diameter	0.30	
	8		28.3		4.3	15.5				0.05	
	9		27.0		4.26	7.0				0.18	
	10		29.5		4.47	5.0				0.30	
Chaudhary, Srivastava [104]	A1	Mix proportion: 1:1.65:3 w/c: 0.46 Slump test 100 mm	26.7	2.25				PET	Low-density PET	0	By weight
	A2		32.7	2.58						0.4	
	A3		35.8	2.67						0.6	
	A4		36	2.64						0.8	
	A5		23.5	2.14						1	
Fraternali, Spadea [105]	CR	Cement: 496 kg Fine agg. 944.1 kg Coarse agg. I 605 kg Coarse agg II 170 kg Water 187.9 kg w/c: 0.38 SP 4.35 kg	33.9					PET	-	-	By total weight
	C0.55L		32.0						1.1 × 40 mm	0.55	
	C0.55S		31.1						0.7 × 52 mm	0.55	
Saikia and de Brito [52]	Ref	Cement: 350 kg Fine agg. 802.7 kg Coarse agg. 996.4 kg Water 185.5 kg	46.3	3.4	4.7	12.7	2378	Crushed PET	-	0	Coarse aggregate replacement by weight
	PC5		33.9	2.4	3.8	12.0	2326		Coarse	5	
	PC10		24.7	1.8	3.0	12.0	2277		Coarse	10	
	PC15		17.2	1.2	2.3	-	2233		Coarse	15	
	PF5		40.6	3.1	4.3	12.2	2336		Fine	5	
	PF10		33.7	2.6	3.7	12.2	2290		Fine	10	
	PF15		29.4	2.2	2.9	12.0	2243		Fine	15	Fine aggregate replacement by weight
	PP5		40.8	3.2	4.5	12.2	2347		Pilled fine	5	
	PP10		39.1	3.1	4.2	12.2	2297		Pilled fine	10	
	PP15		35.2	2.8	3.9	13.2	2254		Pilled fine	15	
Sambhaji [53]	Pl1	Cement: 380 kg Fine agg. 715 kg Coarse agg. 1020 kg w/c: 0.53 kg	44.2		5.9	7.8	2400	Shredded PET	Length 0.15–12 mm and width 0.15–4 mm	0	Fine aggregate replacement by weight
	Pl2		33.2		4.6	2.6	2320			10	
	Pl3		29.4		4.3	1.6	2250			15	
	Pl4		29.8		4.1	0.4	2230			20	
Borg, Baldacchino [106]	Control	Cement: 409 kg Fine agg. 900 kg Coarse agg. 736 kg Water 225 L w/c: 0.55 SP 4.09 kg	28.6		3.55			-	-	0.0	Volume fraction
	S5-0.5		26.2		3.51			Straight PET	50 mm L	0.5	
	S5-1		25.2		4.21			Straight PET	50 mm L	1.0	
	S5-1.5		26.8		4.21			Straight PET	50 mm L	1.5	
	S3-1		27.9		3.94			Straight PET	30 mm L	1.0	
	D5-0.5		27.8		3.71			Deformed PET	50 mm L	0.5	
	D5-1		28.5		4.32			Deformed PET	50 mm L	1.0	
	D5-1.5		27.1		4.00			Deformed PET	50 mm L	1.5	
	D3-1		27.8		4.10			Deformed PET	30 mm L	1.0	
Azhdarpour, Nikoudel [73]	P0	Cement: 10.08 kg Fine agg. I 18.9 kg Fine agg. II 6.3 kg Coarse agg. 25.2 kg Water 0.5	35	2.5	4.4		2160	PET	Crushed	0	Replacing fine aggregate
	P5		51	3.1	6.1		2115			5	
	P10		38	3.3	4.9		2080			10	
	P15		31	2.9	4.8		2050			15	
	P20		29	2.8	4.3		2020			20	
	P25		22	2.2	4.1		1980			25	
	P30		19	1.6	3.0		1930			30	
Islam, Meherier [54]	WC420	Cement: 461.5 kg Fine agg. 534.2 kg Coarse agg. 1024 kg w/c: 0.42	33.4			0.20	2150	Crushed and transformed to aggregate PET		0	Replacing coarse aggregate by weight
	WC422		30.3			1.85	2060			20	
	WC423		27.1			2.00	2037			30	
	WC424		25.9			2.00	2035			40	
	WC425		20.4			0.95	1980			50	
	WC480	Cement: 499 kg Fine agg. 519.8 kg Coarse agg. 996.4 kg w/c: 0.48	32.1			3.1	2145			0	
	WC482		27.6			3.5	2050			20	
	WC483		26.4			3.8	2010			30	
	WC484		24.4			4.0	2000			40	
	WC485		19.4			4.8	1970			50	
	WC570	Cement: 431.6 kg Fine agg. 499.6 kg Coarse agg.: 957.7 kg w/c: 0.57	31.6			10.0	2150				
	WC572		24.2			9.0	2005			0	
	WC573		24.3			10.5	1995			20	
	WC574		22.8			13.1	1985			30	
	WC575		17.4			15.9	1925			40	
Nursyamsi and Zebua [113]	FM601	Cement: 367.27 kg Fine agg. 518.85 kg Coarse agg. 600.88 kg w/c: 0.55	13.8					Shredded and transferred to coarse aggregate PET	Finance modulus	6.0	Replacing by volume coarse aggregate
	FM65		16.2							6.5	
	FM70		16.5							7.0	
Hameed and Fatah Ahmed [74]	A	Mortar	20.6	2.3	6.4		2300	Crushed PET		0	Replacing by volume coarse aggregate
	B	Concrete 0.35 w/c	16.0							0	
	C	Concrete 0. 5 w/c	15.1							0	
	D	Concrete 0.4 w/c	15.4							0	
	E	Mortar	20.7	2.6	4.8		2280			1	
	G	Mortar	17.1	4.1	6.3		2270			3	
	I	Mortar	17.9	4.7	8.8		2180			5	
	K	Mortar	17.5	3.7	8.0		2220			7	
	L	Mortar	16.6	5.5	7.9		2150			10	
Mustafa, Hanafi [55]	Plain	Cement: 400 kg Fine agg. 800 kg Coarse agg. 970 kg	42			16	2210	PET		0	Replacing fine aggregate
	PW5		39			13	2050			5	
	PW10		37			11	1990			10	
	PW20		32			8	1960			20	
Alani, Bunnori [56]	U0	Cement: 1080 kg Fine agg. 760 kg Coarse agg. 380 kg Water 184 L w/c: 0.65 SP 54 kg	134			19.5		PET	40 × 3.5 × 0.3	0	Partial fine aggregate replacement
	U20G		142			21.0				20	
	U40G		140			22.5				40	
	U0P		138			17.0				0	
	U20GP		145			17.5				20	
	U40GP		140			19.0				40	
Gurunandan, Phalgun [27]	CC	Cement: 380.1 kg Fine agg. 859 kg Coarse agg 1095kg Water 152 L SP 4.14 lt Added 0.13% PET Three ratios of shredded rubber were added (7.5%, 15%, and 22.5%)	41.8	3.74	7.00	10	2489	PET polyester fiber	-	0	Adding cement by weight
	RC7		31.8	2.89	6.44	-	-			0.5	
	RC15		24.0	2.34	5.47	-	-			0.5	
	RC22		13.8	1.91	3..00	-	-			0.5	
	FR7		25.9	2.7	5.55	-	-			0.5	
	FR15		19.8	1.88	4.65	-	-			0.5	
	FR22		9.4	1.13	2.80	-	-			0.5	
Almeshal, Tayeh [36]	PET0	Cement: 370 kg Fine agg. 600 kg Coarse agg. 1250 kg w/c: 0.54	28.5	3.11	7.6			-	-	0	Replacing fine aggregate
	PET10		28.2	2.78	7.4					10	
	PET20		27.3	2.51	6.8					20	
	PET30		19.7	2.01	5.9			PET	Crushed	30	
	PET40		11.4	1.74	3.2					40	
	PET50		2.7	0.45	1.2					50	
Hanuseac, Dumitrescu [75]	S0	Cement: 324 kg Fine agg. 803 kg Coarse agg. 558 kg Fly ash: 32.4 kg Water 180 L SP 32.4 kg	33.5	3.9			2260	PET	Chopped	0	Replacing fine aggregate
	S1		23.6	2.1			2100			50	
	S2		20.4	2.2			2000			70	
	S3		14.7	1.9			1900			90	
Mehvish, Ahmed [57]	1-C	Cement: 10 kg Fine agg. 15 kg Coarse agg. 30 kg Water 4.5 L	26.0	2.70	3.10	2.5		PET	20 × 30	0.0	Adding as a ratio of cement weight
	2-0.5%		24.6	2.30	2.90	2.7				0.5	
	3-1.0%		24.3	2.25	2.85	2.8				1.0	
	4-1.5%		24.2	2.10	2.70	3.3				1.5	
Thomas and Moosvi [43]	CS	M50	83	2.6	7.5	9.7		PET fiber	0.25 × 2.3 mm	0.0	Addition
	0FRBC		90	3.6	10	8.9				0.0	
	2FRBC		95	4.3	13	8.5				0.2	
	4FRBC		96	4.7	17	8.1				0.4	
	6FRBC		82	4.5	9	8.0				0.6	
	8FRBC		78	2.4	8	7.4				0.8	
Meena, Surendranath [58]	C251	Cement: 390 kg Fine agg. 835.1 kg Coarse agg. 457.3 kg Water 156.1 L Specific gravity 1.23 Density 1270 kg/m^3^	30.2	24.8		8.3	2520	PET	Aspect ratio 10	0.5	Fine aggregate replacing
	C251		31.3	25.4		7.7	2520		Aspect ratio 10	1.0	
	C251		30.8	24.8		7.5	2510		Aspect ratio 10	1.5	
	C251		29.2	22.7		7.2	2510		Aspect ratio 10	2.0	
	C251		27.0	21.6		7.0	2510		Aspect ratio 10	2.5	
	C251		24.3	18.4		5.7	2510		Aspect ratio 10	3.0	
	C252	Cement: 390 kg Fine agg. 835.1 kg Coarse agg. 457.3 kg Water 156.1 L	31.3	25.7		8.3	2520		Aspect ratio 20	0.5	
	C252		34.1	26.8		6.3	2520		Aspect ratio 20	1.0	
	C252		32.4	26.3		5.3	2520		Aspect ratio 20	1.5	
	C252		30.7	24.6		5.2	2510		Aspect ratio 20	2.0	
	C252		29.0	22.9		4.7	2500		Aspect ratio 20	2.5	
	C252		25.7	19.6		4.3	2490		Aspect ratio 20	3.0	
	C301	Cement: 376 kg Fine agg. 535 kg Coarse agg. 534 kg Water 180.3 L	45.4	35.6		6.7	2540		Aspect ratio 10	0.5	
	C301		47.0	37.3		6.3	2540		Aspect ratio 10	1.0	
	C301		46.1	36.7		5.7	2540		Aspect ratio 10	1.5	
	C301		43.2	34.6		4.7	2530		Aspect ratio 10	2.0	
	C301		37.3	29.7		4.8	2520		Aspect ratio 10	2.5	
	C301		34.6	27.5		4.2	2520		Aspect ratio 10	3.0	
	C302	Cement: 376 kg Fine agg. 535 kg Coarse agg. 801 kg Water 180.3 L	48.6	38		6.7	2540		Aspect ratio 20	0.5	
	C302		48.6	39.1		4.7	2520		Aspect ratio 20	1.0	
	C302		48.4	37.4		3.8	2520		Aspect ratio 20	1.5	
	C302		43.6	34.6		4.0	2510		Aspect ratio 20	2.0	
	C302		40.2	31.6		3.3	2510		Aspect ratio 20	2.5	
	C302		37.4	29		2.7	2510		Aspect ratio 20	3.0	
Liu, Nafees [59]	0SF0	Cement: 367.27 kg Fine agg. 852.73 kg Coarse agg. 928 kg Water 202 L SP 0–14 mL/kg w/c: 0.55 Silica fume 0–73.45 kg	20.5	3.4		8.6	2360	PET		0	Replacing fine aggregate
	1SF2		21.4			8.1	2290			1	
	3SF6		20.7			7.8	2140			3	
	5SF10		20.6			7.5	1990			5	
	7SF14		19.1			7.6	1890			7	
	10SF17		18.1			-	-			10	
	15SF20		16.8			-	-			15	
Steyn, Babafemi [60] 2021	Ref1	Cement: 448 kg Fine agg. 757 kg Coarse agg. 937 kg Water 224 L w/c: 0.5	44.6	4.55		11.3		-		0	Replacing fine aggregate
	Ref2		42.7	4.47		8.5		-		0	
	Pac15		44.6	4.6		8.5		PET		15	
	Pac30		33.1	4.6		7.0		PET		30	
	Rac15		31.7	4.6		7.8		Rubber		15	
	Rac30		22.3	4.6		5.0		Rubber		30	
	Gac15		48.0	4.6		10.2		Glass		15	
	Gac30		45.4	4.6		7.0		Glass		30	
Mohammed and Mohammed [107]	MC	1:1.2:2.4 w/c: 0.5	39.8	3.28	5.89			PET	-	0.0	Volume fraction
	20-0.25		39.9	3.53	5.11				0.44 × 20 mm	0.25	
	35-0.25		37.8	3.28	5.67				0.44 × 35 mm	0.25	
	50-0.25		34.8	3.1	5.57				0.44 × 50 mm	0.25	
	20-0.5		41.2	3.38	6.06				0.44 × 20 mm	0.5	
	35-0.5		38.4	3.37	5.81				0.44 × 35 mm	0.5	
	50-0.5		37.7	3.61	5.92				0.44 × 50 mm	0.5	
	20-1		36.7	3.63	5.61				0.44 × 20 mm	1.0	
	35-1		39.1	3.74	4.45				0.44 × 35 mm	1.0	
	50-1		36.1	3.48	4.31				0.44 × 50 mm	1.0	
	20-0.5		36.3	3.01	5.48				0.11 × 20 mm	0.5	
	35-0.5		33.8	3.18	5.32				0.11 × 35 mm	0.5	
	50-0.5		33.4	3.01	4.64				0.11 × 50 mm	0.5	
Jain, Siddique [108]	A0	Cement: 425.73 kg Fine agg. 653.92 kg Coarse agg. 1177 kg Water 191.6 kg	26.7					Crushed PET		0.0	Adding concrete by weight
	A1		25.9							0.5	
	A2		22.7							1.0	
	A3		15.5							2.0	
	A4		7.1							3.0	
	A5		3.8							5.0	
Meza, Pujadas [61]	Control	Cement: 383 kg Fine agg. 672 kg Coarse agg. 1100 kg w/c: 0.6	31.0	2.50	2.8	4.8		Fibers PET	-	0	Adding concrete by weight
	2-50		30.0	2.30	2.6	3.5			53.5 × 3 × 0.3	2	
	2-110		29.0	2.35	2.7	3.8			117.8 × 3 × 0.3	2	
	6-80		29.5	2.20	2.7	3.8			85.6 × 3 × 0.3	6	
	10-50		29.3	2.25	2.8	3.9			53.5 × 3 × 0.3	10	
	10-110		28.0	2.30	2.9	3.9			117.8 × 3 × 0.3	10	
Singh [62]	1	M40	43.8	3.2	5.4	7.2		Shredded PET	1.18 mm	0	Fine aggregate replacement by weight
	2		44.5	3.4	5.8	6.8				4	
	3		48.6	3.8	6.2	6.5				8	
	4		43.5	3.2	5.6	5.7				12	
	5		40.2	3	5.4	5.2				16	
Tayeh, Almeshal [63]	RCM	Cement: 350 kg Fine agg. 619 kg Coarse agg. 1246 kg w/c: 0.51 kg SP 2.5% for 10% PET		5.5		10.0	2310	Shredded PET		0	Fine aggregate replacement by weight
	PL10			5.3		13.0	2250			10	
	PL20			5.0		16.5	2240			20	
	PL30			4.3		23.0	2210			30	
	PL40			4.0		28.0	2160			40	

## 8. Effects of PET on the Structural Behavior of RC Beams

Table 4 illustrates a list of studies that reused PET as an additive or replacement material in the concrete mix. Additionally, it shows the behavior of reinforced concrete beams when PET is used in the mix as an addition or replacement material. The structural behavior of concrete containing PET was investigated, and ultimate load and deflection were illustrated. Mix design parameters are also listed in the table. Test variables such as PET fraction, aspect ratio, shape, and size are also demonstrated. Finally, the failure mode is illustrated.

Load-carrying capacity is improved when PET is used in the concrete mix. A 10–20% enhancement is observed when 0.5–1.25% PET is added as a fiber addition (Figure 17). The partial aggregate replacement optimum ratio is about 15% for fine or coarse aggregate, as shown in Figure 18.

In terms of deflection, adding PET increases deflection by 20–80% when 0.25–2% is added to the mix, which results in a growth in the member ductility (Figure 19). Some other authors indicate that adding PET would reduce deflection by about 20%. A reduction in deflection and ductility is observed when PET is used as a partial coarse aggregate if the ratio goes beyond 10%, with a non-remarkable enhancement in load-carrying capacity (Figure 20).

## 9. Saving

Researchers started utilizing recycled plastic waste as green, light-weight aggregates to replace, in part or in full, the natural aggregates of concrete. Using PET in concrete structures has led to savings in concrete and steel quantities of up to 7.23% and 7.18%, respectively, depending on the structural configuration of the building [126]. Using PET on several floors of a building could reduce the quantity of concrete by about 5% (Figure 21).

## 10. Conclusions

The increase in plastic waste increases concern about its recycling, its effects on the environment, and its disposal. Hence, researchers conducted studies on utilizing PET in concrete mixtures as an addition or recycling PET as an aggregate replacement. PET affects the mechanical properties of concrete as well as the structural behavior of reinforced concrete beams. The effectiveness increases depending on whether PET is utilized as an additional material or as a replacement material for fine or coarse aggregate. Secondly, it also depends on the ratio of PET. Below are some points that summarize the findings and conclusions:PET can be utilized successfully and effectively to replace traditional fine or coarse aggregate.As the volume ratio of the utilized PET increased, concrete workability decreased.If a concrete mixture with a high ratio of PET is used, water-reducing admixtures are required.The fresh density of concrete containing PET is reduced if PET is added to the mixture. This is due to the low specific gravity of PET compared to the specific gravity of natural fine or coarse aggregate.The permeability of concrete is reduced when a low ratio of PET is used, up to 5%.Compressive strength is increased by about 5% when 0.2–0.4% PET is added to the concrete mixture. Beyond this ratio, compressive strength is gradually reduced.PET polyester fiber can increase compressive strength by 10% to 20% when 0.2 to 0.3% is added.For concrete compressive strength, the optimum PET ratio as a natural aggregate replacement is 1%.The split tensile strength of concrete using PET is remarkably increased by 10–20% when a 0.4–1% PET ratio is used. In the case of using PET as a replacement material, adding 1–8% would increase split tensile strength by 1–20%. On the other hand, if PET is used as a coarse aggregate replacement, that would negatively affect the split tensile strength.In the case of adding PET polyester to the concrete, this leads to a reduction in split tensile strength.Adding PET as an addition material to the concrete mix has no observed enhancement, apart from several authors who claimed different points of view.Adding PET as a replacement for fine aggregate would increase flexural strength by 40% when a 0.5–6% ratio is used. In the case of PET being used as a coarse aggregate replacement, that would negatively affect the flexural strength.Load-carrying capacity is improved when PET is used in the concrete mix. A 10–20% enhancement is observed when 0.5–1.25% is added.Adding 0.25% PET polyester leads to a slight increase in flexure strength of about 6 to 15%.Adding PET increases deflection by 20–40% when 0.25–2% is added to the mix, resulting in growth in member ductility. A reduction in deflection and ductility is observed when PET is used as a partial aggregate replacement, and the ratio goes beyond 10% with a non-remarkable enhancement in load-carrying capacity.Using PET on several floors of a building could reduce the quantity of concrete by about 5%.PET presence enhances cracking performance.

## 11. Future Direction, Gaps, and Recommendations

Utilizing PET in concrete is considered an environmentally friendly method for the disposal of plastic waste. It could also increase the mechanical properties of concrete in some circumstances, and it could affect the mechanical behavior of concrete negatively as well depending on some factors such as the shape of the PET, length, aspect ratio, adding ratio, and concrete strength. Below are some recommendations and future directions for research:Although many studies have investigated the effects of PET length on concrete behavior, the aspect ratio effect is rarely studied.One of the drawbacks of utilizing PET is a reduction in slump test measurement. Therefore, it is recommended to study the effects of different mix designs and additives on increasing workability in PET concrete.Further study is needed on the effects of PET ratio on concrete thermal conductivity and its result on the construction of energy-efficient buildings as environmental concerns.Many studies investigated the effects of different PET ratios on post-cracking without considering the effects of different PET geometry on post-cracking.Further study is needed on the effects of different PET lengths and geometry on split tensile strength.Further study is needed on utilizing a higher PET percentage as a partial fine aggregate replacement without affecting the overall mechanical properties of concrete; the current optimum replacement ratio is 1–5%.Durability is an important aspect and needs further studies looking at abrasion resistance, long-term shrinkage, and creep.The economic evaluation of utilizing PET in concrete needs to be investigated, considering the savings generated by the incorporation of PET as well as the advantages of saving time in the disposal of plastic waste.There has been little consideration for a recycling analysis comparison between traditional plastic waste and recycling PET in concrete.There was a lack of research on modeling concrete using PET.Further study is needed on the effects of using nanomaterials in concrete containing PET.Examine the effects of the PET ratio on water permeability, gas permeability, chloride resistance, and freeze-thaw resistance.Demonstrate the effects of elevated temperatures on concrete containing PET.An experimental study is required to investigate the fatigue and toughness resistance of concrete containing PET.

Through this article, it was possible to demonstrate the main studies that investigated PET as a partial aggregate replacement or used PET as fibers in concrete. Advantages and disadvantages were discussed, in addition to future research directions.

## Figures and Tables

**Figure 1 polymers-15-03320-f001:**
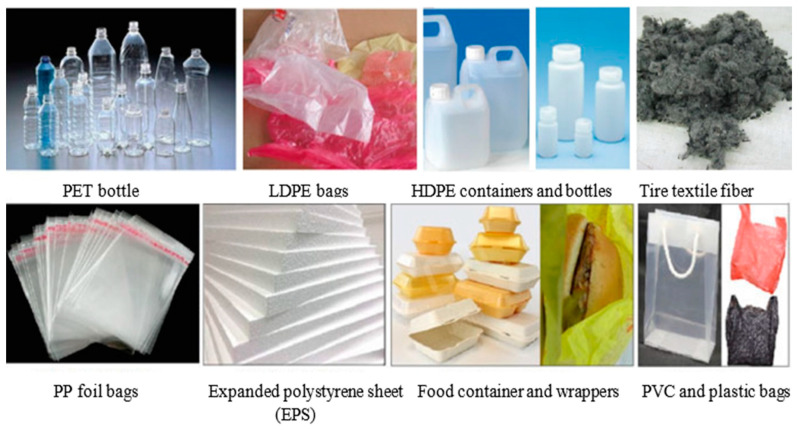
Types of PET waste sources [7]. Adapted with permission from B. A. Mir, Springer, Singapore, 2022.

**Figure 2 polymers-15-03320-f002:**
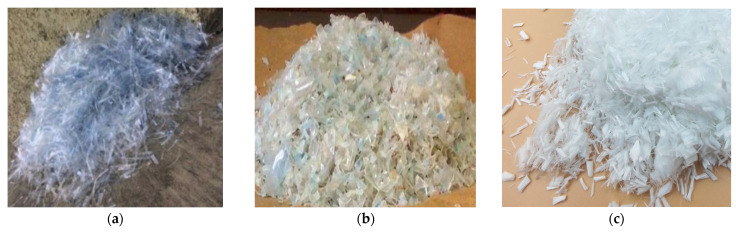
Polyethylene terephthalate (PET) waste fibers. (**a**) PET plastic fiber; (**b**) Shredded PET; (**c**) PET polyester fiber [25].

**Figure 3 polymers-15-03320-f003:**
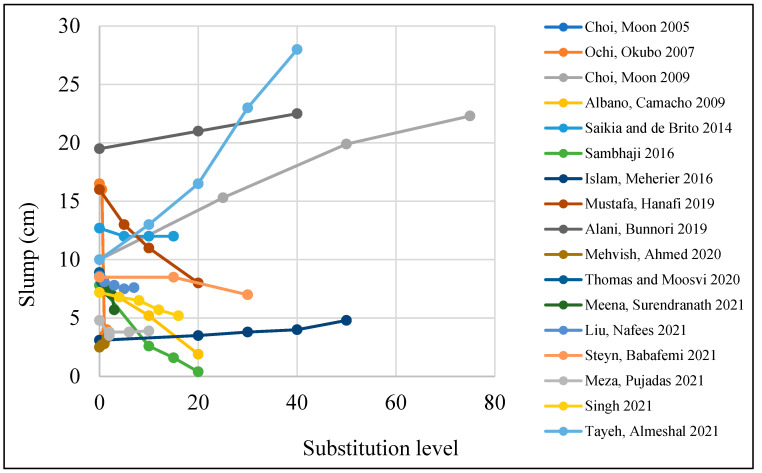
Effects of PET utilization on a slump test [43,48,49,50,51,52,53,54,55,56,57,58,59,60,61,62,63].

**Figure 5 polymers-15-03320-f005:**
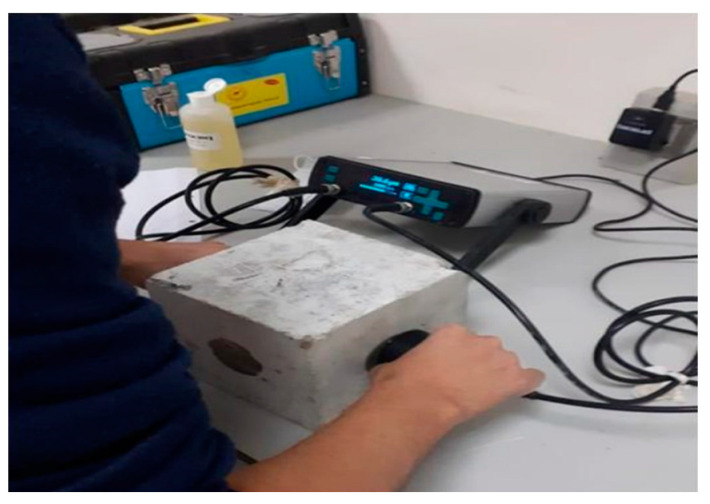
Ultrasonic pulse velocity test [45].

**Figure 6 polymers-15-03320-f006:**
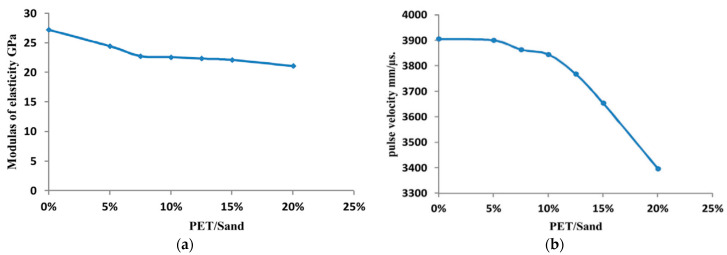
(**a**) Relation between pulse velocity and PET/Sand; (**b**) Relation between modulus of elasticity and PET [45].

**Figure 7 polymers-15-03320-f007:**
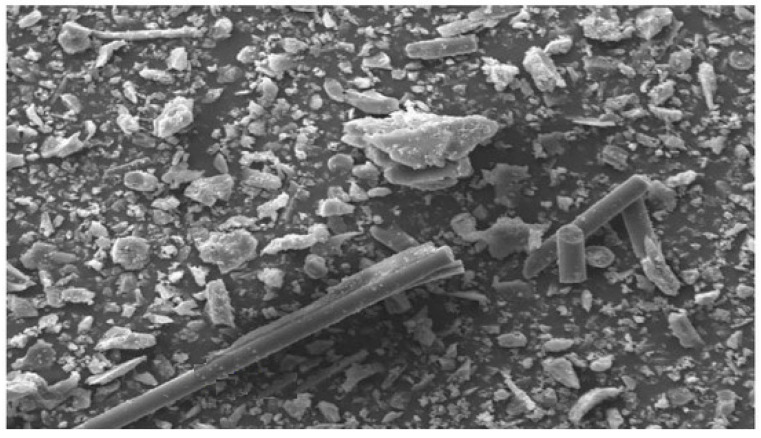
SEM micrographs of samples [95].

**Figure 8 polymers-15-03320-f008:**
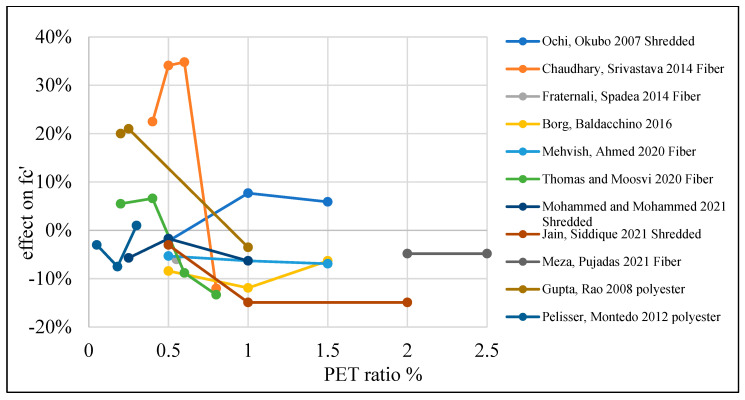
Effects of PET addition on the compressive strength of concrete [26,28,43,49,57,61,104,105,106,107,108].

**Figure 9 polymers-15-03320-f009:**
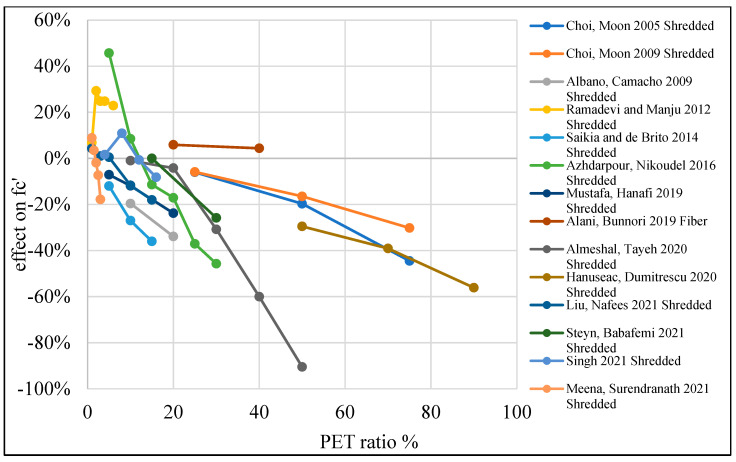
Effects of partial fine aggregate replacement by PET on the compressive strength of concrete [2,48,50,51,52,55,56,58,59,60,62,73,75,109].

**Figure 10 polymers-15-03320-f010:**
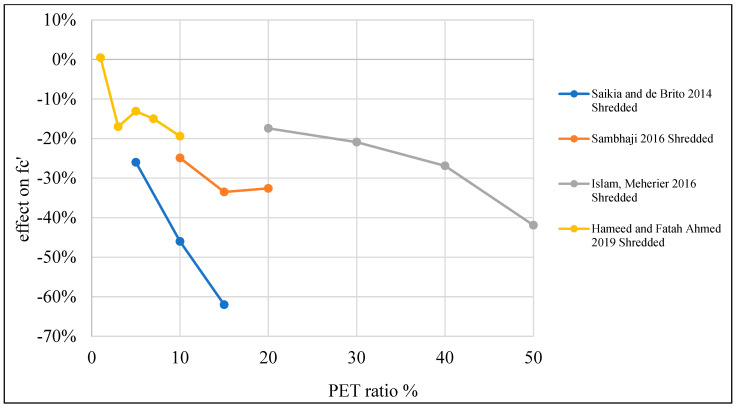
Effects of partial coarse aggregate replacement by PET on the compressive strength of concrete [52,53,54,74].

**Figure 11 polymers-15-03320-f011:**
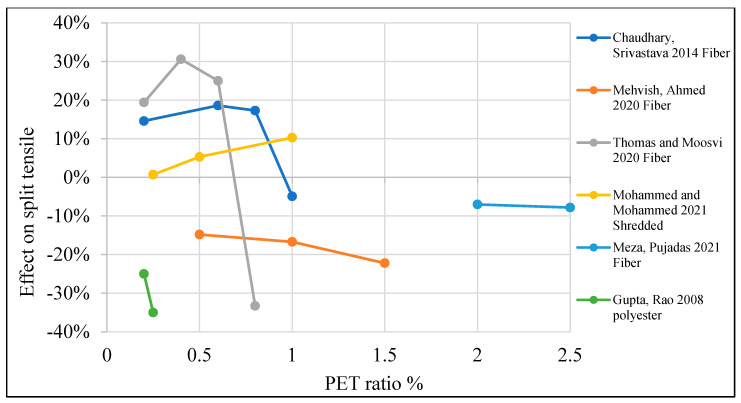
Effects of PET addition on the split tensile strength of concrete [26,43,57,61,104,107].

**Figure 12 polymers-15-03320-f012:**
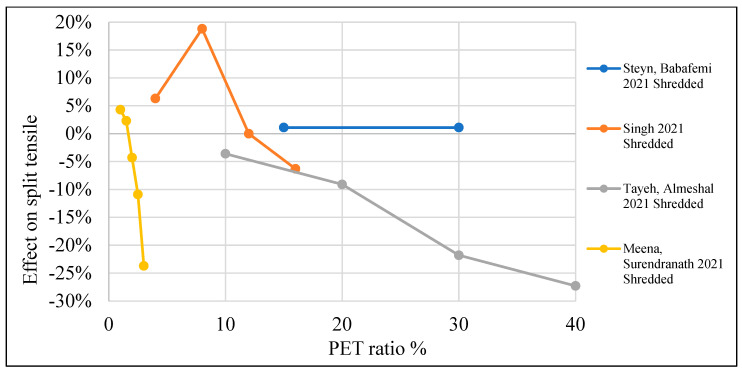
Effects of partial fine aggregate replacement by PET on the split tensile strength of concrete [58,60,62,63].

**Figure 13 polymers-15-03320-f013:**
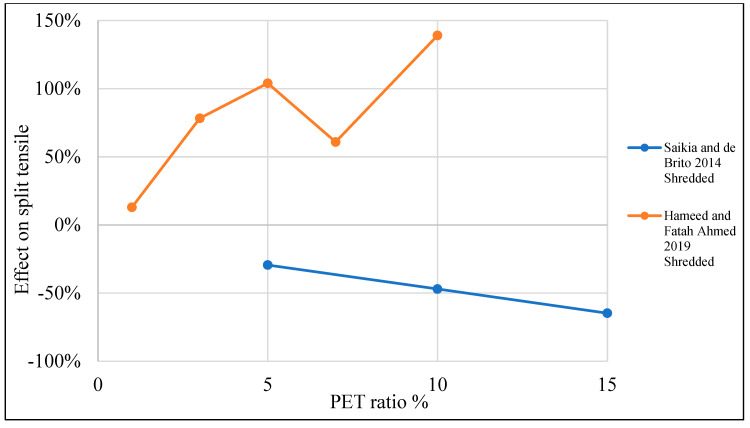
Effects of partial coarse aggregate replacement by PET on the split tensile strength of concrete [52,74].

**Figure 14 polymers-15-03320-f014:**
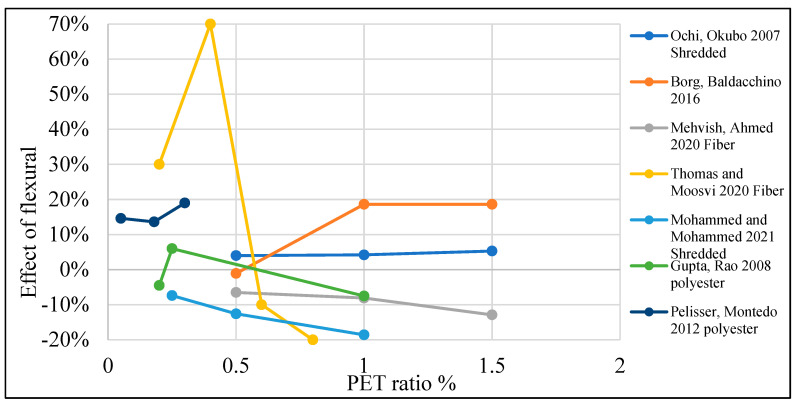
Effects of PET addition on the flexural strength of concrete [26,28,43,49,57,106,107].

**Figure 15 polymers-15-03320-f015:**
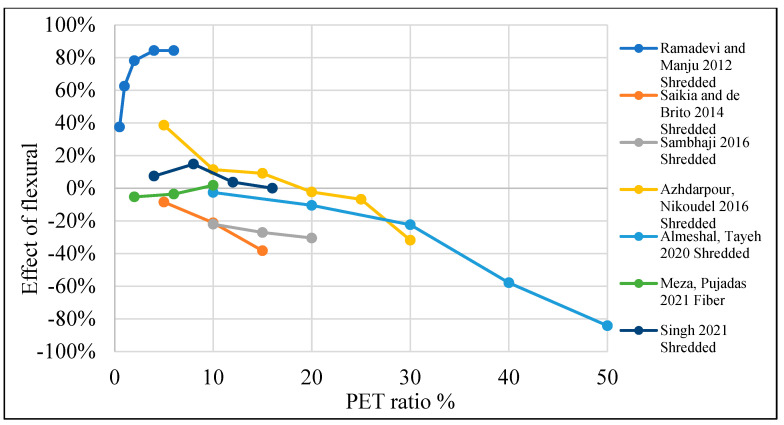
Effects of partial fine aggregate replacement by PET on the flexural strength of concrete [36,52,53,61,62,73,109].

**Figure 16 polymers-15-03320-f016:**
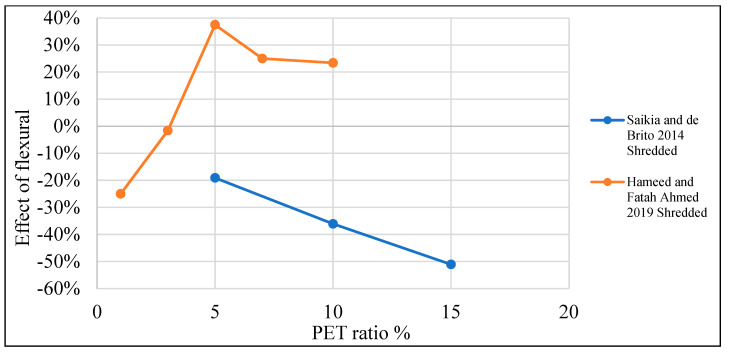
Effects of partial coarse aggregate replacement by PET on the flexural strength of concrete [52,74].

**Figure 17 polymers-15-03320-f017:**
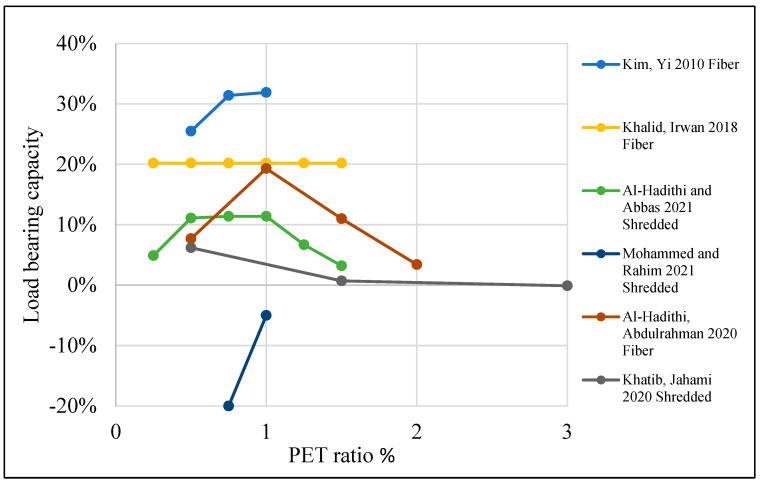
Effects of the PET ratio as an additional fiber on the load-bearing capacity of an RC beam [89,107,114,118,121,123].

**Figure 18 polymers-15-03320-f018:**
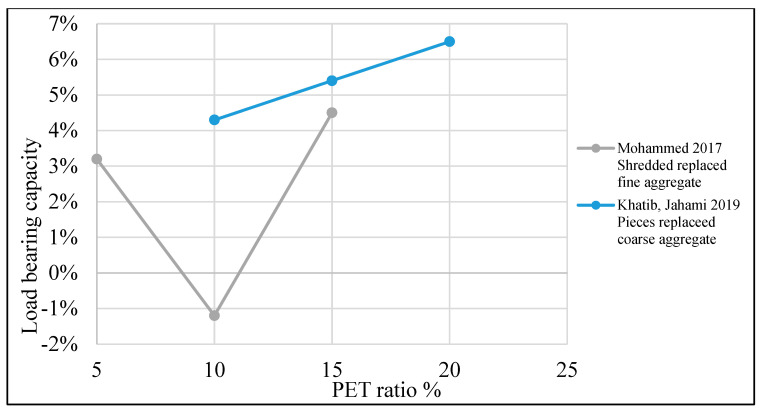
Effects of the PET ratio as a partial aggregate replacement on the load-bearing capacity of an RC beam [116,119].

**Figure 19 polymers-15-03320-f019:**
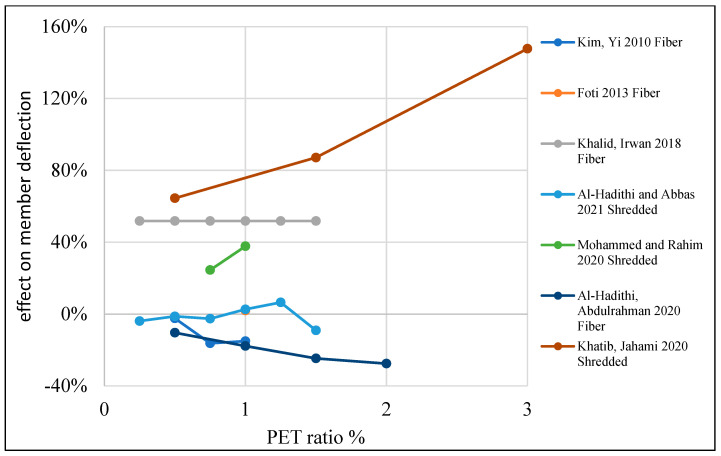
Effects of the PET ratio as an addition on the RC beam deflection. [89,114,115,118,121,122,123].

**Figure 20 polymers-15-03320-f020:**
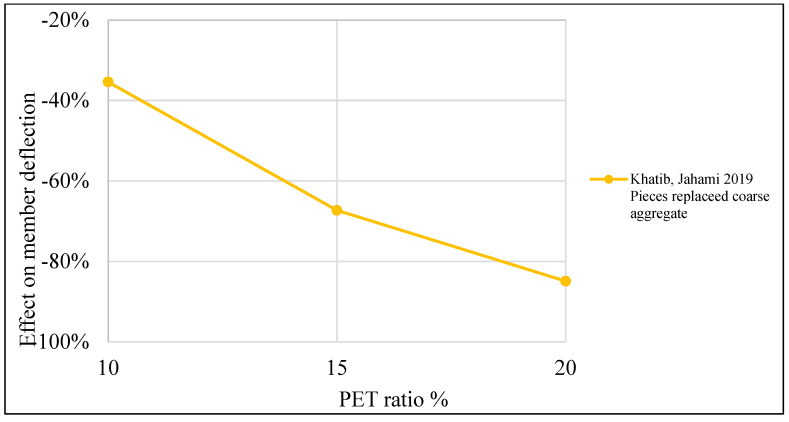
Effects of the PET ratio as a partial aggregate replacement on the RC beam deflection [119].

**Figure 21 polymers-15-03320-f021:**
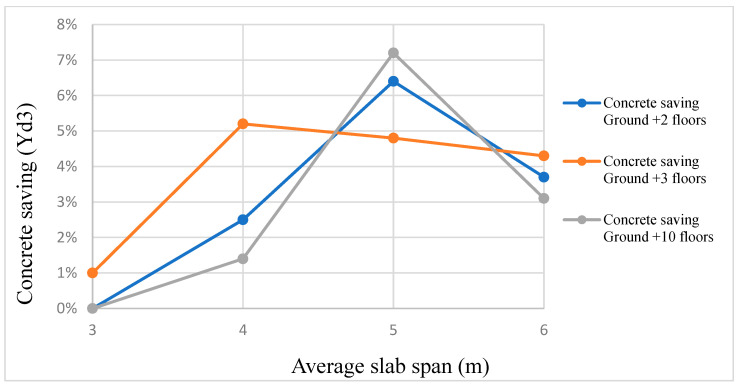
Concrete savings when PET is used, reproduced after [126].

**Table 1 polymers-15-03320-t001:** Properties of recycled plastic and concrete materials [3,12,13].

Material	f_t_ (MPa)	E (GPa)	λ (W/m.k)	Specific Gravity
PET	55–80	2.1–3.1	0.15	1.3–1.4
PVC	50–60	2.7–3.0	0.17–0.21	1.3–1.4
PS	30–55	3.1–3.3	0.105	1–1.1
PP	25–40	1.3–1.8	0.12	0.9–0.91
PE	18–30	0.6–1.4	0.33–0.52	1.2–1.28
Aggregate	-	70	2.29–2.78	2.55–2.65
sand	-	70	4.45	2.6–2.7
Cement paste (w/c = 0.5)	2.5–4.0	36–40	1	3.1–3.15

**Table 2 polymers-15-03320-t002:** Water absorption (%) of partial fine aggregate replacement adapted from [79,80,81]. Reproduced from Laurent Molez, Elsevier, 2015; Bartolomeo Coppola, Elsevier, 2018; Abu Hasan, DUET, 2015.

Plastic Fiber (%)	0.0	5.0	10	15	20	25	50
Ezziane et al., 2015 [79]	2.2	2.2	2.4	4.8			
Coppola et al., 2018 [80]	7.2	7.2	7.4	7.2	7.2	7.6	8.0
Hassan et al., 2015 [81]	8.0	8.2	8.2	9.4	9.5	9.8	18.3

**Table 4 polymers-15-03320-t004:** Studies that utilized PET in structural concrete.

Author	Beam ID	Beam DimensionB × H × L (cm)	Concrete Strength (MPa)	Fc’ (MPa)	Ft (MPa)	Sample Parameter/Remarks	Material Types	Dimension (mm)	Ratio % V	Ultimate Load (kN)	Ultimate Deflection (mm)	Failure Mode
Kim, Yi [114]	NF	10 × 10 ×40	**	26		994 kg/m^3^ coarse agg.	PET	-	-	121.6	169	Flexural
	RPET 0.5			26		775 kg/m^3^ fine agg.		0.2 × 1.3 × 50	0.5	152.6	165	
	RPET 0.75			25		355 kg/m^3^ cement		0.2 × 1.3 × 50	0.75	159.8	141.4	
	RPET 1.0			24		161 kg/m^3^ water		0.2 × 1.3 × 50	1	160.4	143.4	
	PP 0.5			26		40 kg/m^3^ fly ash		0.38 × 0.9 × 50	0.5	154	140.1	
	PP 0.75			24.5		2.37 kg/m^3^ air entainer		0.38 × 0.9 × 50	0.75	150.4	149.2	
	PP 1.0			24		w/c 0.41 Sand/Aggregate 43.8%			1	156.6	144.2	
Foti [115]	B1	10 ×10 × 40	**	53.2	2.34	PET 0.5–0.75% (0.0%w) superplasticizer PET 1% (0.8% w) superplasticizer PET little beam (1.4% w) superplasticizer	PET	circular PET	-	4	20	Flexural
	B2							half bottle PET	1W	4.6	20	
								circular + 2				
	B3							overlaped half	1W	3.1	20.4	
								half bottle + 2				
	B4							overlaped half	1W	3.1	20.4	
	B5	10 × 20 × 110		51.5	2.3			circular + 4 layer overlaped half half bottle + 4 layer overlaped half	1W	11	-	
	B6								1W	11	-	
Mohammed [116]	CH100		**	33.1		Concrete mix 1:1.25:2.5 w/c 0.5 Flexure-critical Shredded PET replacing fine aggregate	-	-	-	40.4		Flexural
	PET510			27.1			Shredded	<12.5	5	41.7		
	PET1100			31.8			PET	<12.5	10	39.9		
	PET1510			32.6				<12.5	15	42.2		
	CH200			31.4			-	-	-	112.8		
	PET520			23.8			Shredded	<12.5	5	105.1		
	PET1200			24.9			PET	<12.5	10	100.1		
	PET1520			23.7				<12.5	15	96		
Thomas and Faisal [117]	Bc	10 × 10 × 50	**	25		mix 1:1.45:2.68 w/c 0.45	-	-		13		Flexural
	BPET-mesh						PET mesh	10 × 0.5		8.5		
Khalid, Irwan [118]	B-normal	15 × 30 × 250	**	34.1	0.15 fr	Vf		-	0	98.5	43.1	
	B-RPET-5			34.5	0.22 fr			ring RPET-5 width	0.25	99.3	43.3	
				35					0.5			
				35.3					0.75			
				34.5					1			
				34.8					1.25			
				35.3					1.5			
	B-RPET-10			34.5	0.23 fr			ring RPET-10 width	0.25	98.3	54.4	
				35					0.5			
				35.3					0.75			
				34.5					1			
				34.8					1.25			
				35.3					1.5			
	B-IRE PET			34.1	0.19 fr			PET irregular	0.25	98.3	51.8	
				34.9					0.5			
				34					0.75			
				35.1					1			
				34.7					1.25			
				34.3					1.5			
	B-WRE			33.9	0.31 fr			Waste wire 55 mm	0.25	98.3	53.7	
				34.2					0.5			
				35.3					0.75			
				35					1			
				34.9					1.25			
				34.8					1.5			
	B-SYNT			34.2	0.22 fr			Synthetic fibers	0.25	103.2	57.9	
				34.5					0.5			
				34.4					0.75			
				34.2					1			
				34.8					1.25			
				34.9					1.5			
Khatib, Jahami [119]	PBC 0	20 × 30 × 120	*	15		942.7 kg/m^3^ coarse agg. 942.7 kg/m^3^ fine agg. 314 kg/m^3^ cement 188.5 kg/m^3^ water Replacing coarse aggregate		-	0	92	120	Flexural
	PBC 10			16				PP waste cap	10	96	77.5	
	PBC 15			17.5				PP waste cap	15	97	39.2	
	PBC 20			18.5				PP waste cap	20	98	18.1	
Dawood and Adnan [120]	B1-S	15 × 20 × 140	**	35.8	3.1 fr	1024 kg/m^3^ coarse agg. 649.644 kg/m^3^ fine agg. 95.12 kg/m^3^ cement 201.38 kg/m^3^ water 3.961 L/m^3^ superplasticizer w/c 0.41 Replacing main reinforcement		Steel bar		82.5	12.7	Flexural
	B2							No reo		30	4	
	B3-P1							Plastic bar 1		12.5	16	
	B4-P2							Plastic bar 2		15	17	
	B5-P3							Plastic bar 3		15	17	
	B6-P4							Plastic bar 4		20	20	
	B7-P5							Plastic bar 5		20	16	
	B8-P6							Plastic bar 6 + steel		85	27	
	B9-P7							Plastic bar 7 + steel		25	30	
	B10-P8							Plastic bar 8 + steel		30	29	
	B11-P9							Plastic bar 9 + steel		30	28	
	B12-P10							Plastic bar 10		15	16	
Al-Hadithi and Abbas [121]	Group A	10 × 15 × 100	**	32.9	2.93	Shear-critical beams Steel shear reinforcement	ShreddedPET	40 × 4 × 0.35	0	142.6	7.7	Shear /Flexural shear
				33	3.06				0.25	143.1	7.4	
				33.3	3.07				0.5	142.3	7.6	
				34.6	3.18				0.75	150.1	7.5	
				35.3	3.33				1	154.8	7.9	
				32	3.47				1.25	147.5	8.2	
				32	3.56				1.5	134.2	7	
	Group B			32.9	2.93	CFRP sheet shear reinforcement	ShreddedPET	40 × 4 × 0.35	0	139.8	8.4	
				33	3.06				0.25	146.7	8.1	
				33.3	3.07				0.5	155.3	9.9	
				34.6	3.18				0.75	155.8	10.7	
				35.3	3.33				1	155.8	9.4	
				32	3.47				1.25	149.2	8.6	
				32	3.56				1.5	144.3	7.6	
Mohammed and Rahim [122]	Bc	12 × 15 × 120	***	94.3	4.36	1075 kg/m^3^ coarse agg. 677.5 kg/m^3^ fine agg. 480 kg/m^3^ cement 79.9 kg/m^3^ water 104 kg/m^3^ silica fume 4.16 kg/m^3^ superplasticizer PET specific gravity 1.4	-	-	0	62.4	14.8	Flexural
	B-0.75-S			84.7	3.95		ShreddedPET	1.4 × 20	0.75	47.9	16.5	
	B-0.75-H			77.3	4.2			1.4 × 20	0.75	63.5	18.1	
	B-0.75-L			66.2	4.06			1.4 × 40	0.75	51.9	21.1	
	B-1-S			68.4	3.87			1.4 × 20	1	59.6	20.4	
	B-1-H			68.7	3.62			mixed	1	59.1	20.4	
Adnan and Dawood [25]	Bcr	15 × 20 × 140	**	30.3	4.53 fr	1024 kg/m^3^ coarse agg. 649.644 kg/m^3^ fine agg. 496 kg/m^3^ cement 201.38 kg/m^3^ water 3.961 L/m^3^ superplasticizer water/cement ratio of 0.41	-	-	-			Flexural
	B1			31	4.25 fr		Machine PET	<25.4	1.5	82	12.6	
	B2			30.8	4.33 fr		Machine PET	<25.4	3	75	20	
	B3			43.1	4.91 fr		Hand PET	4 × 40	1.5	72	15	
	B4			24.9	4.31 fr		Hand PET	4 × 40	3	70	25	
Al-Hadithi, Abdulrahman [123]	M1-26s	10 × 15 × 110	**	32.1		Specific gravity 1.12 Mix design 1:1.5:3.15 w/c 0.43	PET	4 × 30 × 0.3	0	86.1	15.2	Flexural
	M2-58As			32.1					0	65.8	13.3	
	M3-6s			32.1					0	29	9	
	M4-26s			33.7					0.5	92.7	14.4	
	M5-58s			33.7					0.5	72.5	12.5	
	M6-6s			33.7					0.5	35	7.2	
	M7-26s			35.5					1	102.7	13.3	
	M8-58s			35.5					1	81.3	11.1	
	M9-6s			35.5					1	38.1	6.8	
	M10-26s			34.6					1.5	95.6	12.3	
	M11-58s			34.6					1.5	75.8	9.8	
	M12-6s			34.6					1.5	35.3	6.4	
	M13-26s			33.3					2	89	11.9	
	M14-58s			33.3					2	69.7	9.6	
	M15-6s			33.3					2	33	6	
Khatib, Jahami [89]	PS-0.0	20 × 30 × 150	**	38.7	4.19	1340 kg/m^3^ coarse agg. 670 kg/m^3^ fine agg. 670 kg/m^3^ cement 270 kg/m^3^ water Mix design 1:1:2 w/c 0.4 Shredded waste plastic PP	-	-	0	181.4	15.5	Flexural
	PS-0.5			40	4.28		PP shredded	2 × 30	0.5	192.7	25.5	
	PS-1.5			36.5	4.33		PP shredded	2 × 30	1.5	182.7	29	
	PS-3.0			36	4.47		PP shredded	2 × 30	3	181.3	38.4	

*: low-strength concrete less than 20 MPa; **: normal-strength concrete 20–54 MPa; ***: High-strength concrete 55–149 MPa [124,125].

## Data Availability

Data are available on request.

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
