# Peer review of "Utilizing Polyethylene Terephthalate PET in Concrete: A Review"

_polymers, 2023, doi:10.3390/polym15153320_

Round 1
Reviewer 1 Report
The paper reviewed the application status of waste plastic PET in concrete materials. I think that this study is generally well written. It seems the authors conducted an extensive literature survey while carefully and thoroughly elaborating the subject. In my opinion, it can be published, after revision based on reviewers’ comments.
1. The information in the tables should be simplified. For example, in table 3, the information of each sample was listed, making the table is too informative. The authors should select optimal sample to explain their opinion.
2. I think the section 1-4 can be integrated and simplified.
3. I suggest that the effect of PET on the microstructural feature and evolution of concrete should be added, making the structure of the paper more complete.
Author Response
We are grateful for the editor and referee’s comments (in blue) and have taken them into account (in red) as follows:
Reviewer#1:
The paper reviewed the application status of waste plastic PET in concrete materials. I think that this study is generally well written. It seems the authors conducted an extensive literature survey while carefully and thoroughly elaborating the subject. In my opinion, it can be published, after revision based on reviewers’ comments.
We appreciate your nice comments on our article, they certainly allowed us to improve our paper.
- The information in the tables should be simplified. For example, in table 3, the information of each sample was listed, making the table is too informative. The authors should select optimal sample to explain their opinion.
Resolved.
We appreciate this point. The author tried to collect as much information on this subject in a table. All tables in the manuscript have been rearranged any all necessary changes were implemented.
- I think the section 1-4 can be integrated and simplified.
Resolved.
Your suggestions are significant and made our manuscript better. The authors have undertaken this into account and corrections were made on sections 1 to 4.
- I suggest that the effect of PET on the microstructural feature and evolution of concrete should be added, making the structure of the paper more complete.
Resolved. Section 7.6 was added to the manuscript and five recent articles were cited (page 12)
Section 7.6: Effects of PET on microstructure of concrete
To investigate the microstructure of concrete, a scanning electron microscope (SEM) is usually used. concrete containing PET show relatively irregular form. It leads to formation of pores of about 2–4 µm, multiple bright inclusions (cement formations) encircled by hydrating agents could observed on the surface which leads to improve the bonding between the PET fibers and the matrix Figure 8. Concrete containing PET probably has much denser interface between the PET aggregates and the cement matrix. Moreover, Microcracks reduce with the present of PET fibers [83, 84]. Aslani 2019 [85] and Hou 2019 [86] reported that the compressive strength decreases with the addition of plastic fibers. Furthermore Aslani 2019 [85] found that increasing the volume fraction of plastic fibers from 0.1 % to 0.2 %, would decreases the compressive strength by about 20 %.
On the other hand Faraj 2020 [86] claimed that, concrete microstructure show improvement in compressive strength be due to the distribution of the fibers inside the microstructure of concrete. This leads to the reduction in the pores inside the concrete matrix. Length of the fibers has a slight influence on the compressive strength of concrete [87].

Reviewer 2 Report
The manuscript "Utilizing Polyethylene Terephthalate PET in Concrete: A Review" needs revision as introduction and some sections starting quite well but there are some parts quite confusing.
1. Did most figures the authors made by them self and if not a source should be given.
2. A general question hence PET is a thermoplastic, formation in concrete does such not change its form and some part of degradation of PET taking place?
3. PET in landfill can endure 500 years now if its in concrete the waste still there as most building do not extent 100 years. So the purpose removing PET from environment is only limited.
4. The tables given on end are very chaotic also no discussion given of the values showing there. There need to be some actual description what those value means.
5. The review need a bit better organised with some examples from former research given as the figures you show are less informative. Please make this review in a more scientific meaning deep work.
Author Response
Manuscript Number: polymers-1947904
Title: Utilizing Polyethylene Terephthalate PET in Concrete: A Review
We are grateful for the editor and referee’s comments (in blue) and have taken them into account (in red) as follows:
Reviewer #2:
The manuscript "Utilizing Polyethylene Terephthalate PET in Concrete: A Review" needs revision as introduction and some sections starting quite well but there are some parts quite confusing.
I sincerely appreciate your review. Your insight was valuable and improved our manuscript.
Did most figures the authors made by them self and if not a source should be given.
Resolved. We appreciate this point. References were in-text cited as shown in all figures’ caption. That is totally right, we have made all the Tables (1-4) Figures (4-5 & 8-21) in the Manuscript.
- A general question hence PET is a thermoplastic, formation in concrete does such not change its form and some part of degradation of PET taking place?
Resolved.
PET is a thermoplastic resin that is particularly well suited for the manufacturing of many products due to its transparency, resilience, and gas barrier capabilities. It is projected that it will take several decades for PET plastic water bottles, a common type of plastic, to decompose completely. It has been documented in the literature that the addition of plastic aggregate to concrete reduces its strength and flexural ratio because both substances are not interconnected properly. Because larger PET particles occupy less space in the concrete, Young's modulus decreases when PET particle size is raised. As a result, compressive strength and compression strain increase. When less PET size is used in the concrete mix, deformations tend to be smaller, but maximum stresses have larger magnitudes. However, waste PET strip concrete exhibits higher loading capacity, delay in damage propagation, lower variability, and better damage tolerance under cyclic loading.
- PET in landfill can endure 500 years now if its in concrete the waste still there as most building do not extent 100 years. So the purpose removing PET from environment is only limited.
Resolved. Thank you for bringing this up. Buildings may also be recycled after demolition as another option for recycling PET. The second issue is that this topic has been the subject of several studies, which is why we, the authors, conducted this study.
- The tables given on end are very chaotic also no discussion given of the values showing there. There needs to be some actual description what that value means.
Resolved. The authors of the Manuscript appreciate you bringing this up. Nearly all of the Figures' discussions have been adjusted, and Figures 4-5 and 8-21 have all been redrawn.
- The review needs a bit better organized with some examples from former research given as the figures you show are less informative. Please make this review more scientific meaning deep work.
Resolved. I appreciate you mentioning this. As an additional alternative, the whole manuscript has been rearranged and some information was relocated, Table and Figures were enhanced in the revised version.
Round 2
Reviewer 2 Report
The authors made revision and the manuscript now in publishable form.
Author Response
Dear reviewer, we appreciate your time and your efforts which made our manuscript better for sure.